# Research

ecology, evolution, microbiology

adaptive landscape, bacteriophages, byproduct evolution, cryptic genetic variation, pleiotropy, phenotypic plasticity

**Author for correspondence:**
Lisa Freund
e-mail: lisa.freund@env.ethz.ch

# Hidden paths to endless forms most wonderful: parasite-blind diversification of host quality

Lisa Freund, Marie Vasse and Gregory J. Velicer

Institute for Integrative Biology, ETH Zürich 8092, Zürich, Switzerland

  LF, 0000-0001-8235-2015; MV, 0000-0003-4081-6664; GJV, 0000-0001-8721-1502

Evolutionary diversification can occur in allopatry or sympatry, can be driven by selection or unselected, and can be phenotypically manifested immediately or remain latent until manifested in a newly encountered environment. Diversification of host–parasite interactions is frequently studied in the context of intrinsically selective coevolution, but the potential for host–parasite interaction phenotypes to diversify latently during parasite-blind host evolution is rarely considered. Here, we use a social bacterium experimentally adapted to several environments in the absence of phage to analyse allopatric diversification of host quality—the degree to which a host population supports a viral epidemic. Phage-blind evolution reduced host quality overall, with some bacteria becoming completely resistant to growth suppression by phage. Selective-environment differences generated only mild divergence in host quality. However, selective environments nonetheless played a major role in shaping evolution by determining the degree of stochastic diversification among replicate populations within treatments. Ancestral motility genotype was also found to strongly shape patterns of latent host-quality evolution and diversification. These outcomes show that (i) adaptive landscapes can differ in how they constrain stochastic diversification of a latent phenotype and (ii) major effects of selection on biological diversification can be missed by focusing on trait means. Collectively, our findings suggest that latent-phenotype evolution should inform host–parasite evolution theory and that diversification should be conceived broadly to include latent phenotypes.

## 1. Introduction

Wonderful 'endless forms' of phenotypes [1–3] often first evolve non-adaptively [4–7], even if they later prove beneficial in a new context as 'exaptations' [8]. Temporally, a non-adaptively evolved phenotype might be generated immediately when its causal genotype first evolves, or only later, upon that genotype's exposure to a novel or changed environment. Here, we refer to a phenotype that is potentiated by an existing genotype but not initially manifested owing to environmental specificity as a 'latent phenotype'. We additionally refer to the evolution of such a genotype prior to the manifestation of its initially latent phenotype as 'latent-phenotype evolution' (or 'LPE', [7]), a label independent of the particular causes or consequences of LPE (see Methods). LPE is intrinsically non-adaptive because the focal phenotype is not yet generated when its genetic basis first evolves. However, the genetic basis of the latent phenotype might evolve by any evolutionary mechanism (table 1). For example, the causal genotype might arise adaptively owing to one phenotypic effect beneficial in a first environment while pleiotropically potentiating a second phenotypic effect that is only manifested in a distinct environment encountered later [9,10]. Alternatively, genotypes underlying LPE may evolve non-adaptively by hitchhiking with an adaptive genetic element [11] or by stochastic forces [6,12]. Variation among initially neutral alleles underlying latent phenotypes [12,13] has long

**Table 1.** Categories of latent-phenotype evolution (LPE). (The genetic basis of a focal latent phenotype first evolves in a temporally prior environment $E_P$. We distinguish three mechanisms responsible for evolution of the causal genotype: adaptation, hitchhiking and stochasticity (corresponding to hypothetical alleles $a$, $h$ and $s$, respectively). By definition, each allele potentiates a latent phenotype $LP_y$ (right superscript) in the prior environment $E_P$ that is then manifested in a later-encountered environment $E_L$ (manifested phenotype, $MP_y$). In addition, this allele may cause a non-advantageous manifested phenotype $MP_x$ (left superscript) in the prior environment $E_P$ that might or might not also be manifested in $E_L$ (as shown by the use of the brackets). All phenotypes manifested in $E_L$ can have positive, negative or no fitness effect. The first (non-header) row of the table describes a form of pleiotropy—a delayed, environment-contingent pleiotropic effect of an adaptive allele. The rows below describe scenarios consistent with the common meaning of 'cryptic genetic variation' (see Methods).)

| prior environment $E_P$ | later environment $E_L$ | cause of allele's increase in $E_P$ | allele's adaptive status in $E_P$ |
|---|---|---|---|
| ${}^{MPx}a^{LPy}$ | ${}^{(MPx)}a^{MPy}$ | adaptation (direct selection) | adaptive |
| $h^{LPy}$ | $h^{MPy}$ | hitchhiking (indirect selection) | non-adaptive |
| ${}^{MPx}h^{LPy}$ | ${}^{(MPx)}h^{MPy}$ | hitchhiking (indirect selection) | non-adaptive |
| $s^{LPy}$ | $s^{MPy}$ | stochasticity (mutation/drift) | non-adaptive |
| ${}^{MPx}s^{LPy}$ | ${}^{(MPx)}s^{MPy}$ | stochasticity (mutation/drift) | non-adaptive |

been recognized as potential fuel for later adaptation to new and changing environments [14–16].

Latently evolved phenotypes can be features of individual organisms. For example, bacteria have latently evolved altered antibiotic resistance [10,17], metabolic-profile shifts [15] and changes in nutrient-uptake abilities [18]. However, outcomes of interactions between organisms can also be considered phenotypes. Examples of such 'interaction phenotypes' [19,20] include reproductive incompatibility resulting from allopatric speciation, which remains latent until allopatrically diverged lineages make secondary contact [21–23]. Similarly, host–parasite interaction phenotypes [24–28] and bacterial social–interaction phenotypes [29,30] can also evolve and diversify latently in allopatry. Given that (i) pleiotropy, hitchhiking and genetic drift are common, (ii) manifestation of phenotypes is often context-specific (e.g. owing to phenotypic plasticity [31] or, for interaction phenotypes, limitations on interaction opportunities), (iii) exposure to changing or new environments is inevitable for most biological lineages, and (iv) latently evolved phenotypes often have selective significance upon their manifestation, LPE is likely to strongly contribute to long-term patterns of phenotypic evolution and diversification.

Bacteria engage in a vast array of interactions, including with their own viral parasites, bacteriophages. Bacteria–phage interactions are determined by the match between phage-infectivity and bacterial resistance mechanisms, which can result in narrow to broad host ranges [32–34]. Diverse mechanisms to resist phage have evolved at all major stages of the infection cycle, including preventing phage adsorption, impeding post-entry reproduction and assembly, and stopping virion release through abortive systems that kill both phage and host [32,35]. Selection to resist infection can lead to host–phage incompatibility, as antagonistic coevolution between phage and their hosts leads to rapid local adaptation [36,37] and diversification [27,38,39]. However, how much host quality—the degree to which a host genotype or population facilitates parasite growth—is shaped directly by parasite-imposed selection versus indirectly from byproducts of other selective forces (e.g. resource competition [26] or other predators [40]) or stochastic forces has been little investigated.

Perhaps all microbes express some social traits [41], but some have evolved extraordinarily complex suites of cooperative behaviours [42]. Myxobacteria, including the model species *Myxococcus xanthus*, engage in cooperative swarming [43] during group predation [44] and in multicellular fruiting-body development [45]. As predators of many microbes, myxobacteria are predicted to strongly shape the structure and evolution of soil microbial communities [46,47]. Myxobacteria are themselves subject to selective pressure by myxophages [48], which in turn are likely to strongly shape myxobacterial social evolution. For example, cell-surface molecules such as type-IV pili and O-antigen serve as phage receptors in many bacteria [49,50] and also function in *M. xanthus* social behaviours [51,52]. Thus, just as bacteria–phage coevolution can indirectly shape bacterial social interactions [53–55], social evolution in the absence of phage is likely to latently alter the character and diversity of future host–parasite interactions. Analyses of experimentally evolved lineages [29,30] suggest that intra-specific social interactions between natural *M. xanthus* lineages [56] often evolve latently. But how LPE shapes future antagonistic interactions of myxobacteria with other species, including with phage and their own prey, remains unexplored.

In this study, we test for LPE—including diversification—of host-virus interactions using the virulent myxophage Mx1 [57] and bacterial populations from an evolution experiment named MyxoEE-3 (see Methods). In MyxoEE-3, populations of *M. xanthus* underwent selection for increasing fitness at the leading edge of colonies expanding spatially by growth and motility, with cells near the leading edge transferred to a fresh plate at the end of each evolutionary cycle [29,30,58]. Importantly, evolving populations never encountered phage during MyxoEE-3, thereby allowing us to test whether and how phage-blind adaptation to multiple environments indirectly shapes the character of interactions with obligate parasites.

We first analyse host–phage LPE in eight MyxoEE-3 treatments that share a common ancestor but differed in their selective environments with respect to surface structure (hard agar (HA) or soft agar (SA)), nutrient availability (high- or low-nutrient levels) and/or nutrient source (nutrient medium alone or with prey lawns). We examine effects of environment and chance on the direction and degree of average phage-blind host-quality evolution and also on the degree of within-treatment diversification. If LPE is mediated predominantly by alleles that increased because of selection during MyxoEE-3, adaptive landscape structure [59,60] may shape LPE outcomes, including the degree to which latent phenotypes diversify stochastically. We further analyse how ancestral motility genotype shapes LPE by testing for effects of debilitating each of the two *M. xanthus* motility systems on latent host-quality evolution and diversification.

## 2. Methods

### (a) Semantics and nomenclature

#### (i) On use of 'latent-phenotype evolution'

Large bodies of literature examine modes by which the genetic basis of latent phenotypes evolves, for example as initially adaptive alleles that potentiate initially unrealized pleiotropic phenotypes [9,61,62] or as initially neutral alleles, variation for which is commonly referred to as 'cryptic genetic variation' (CGV) [12,13]. However, there does not appear to be a well-established generic label for the evolution of the genetic basis of latent phenotypes that is impartial with regard to causes or consequences of such evolution and that applies equally to the phenotypes of individuals and phenotypes of interactions between organisms.

We adopt 'latent' [63] over 'cryptic' owing to the frequent association of the latter with selectively neutral alleles [12,13,16] (despite exceptional applications to initially adaptive alleles [64]) and the evocation of future manifestation by 'latent'. We use 'latent-phenotype evolution' to focus primary attention on the process of evolution over time, which may result in the loss of variation owing to fixation of an allele underlying a latent phenotype, rather than primarily on within-population variation at loci encoding such alleles, which is the focus of CGV. We also note that LPE in our sense is distinct from (but may nonetheless be related to) the latent evolvability of a genotype, population or species, i.e. the potential for the future evolution of novel forms, functions or diversity [65–68].

#### (ii) On use of 'manifestation'

Here, we use 'manifestation' (and variations thereof) rather than 'expression' to refer to the actualization of a genetically caused phenotype. This is because we conceive expression to be the actualization of a phenotype by an individual organism, but we desire a term that also applies generically to organism-interaction phenotypes. Actualization of the latter may be prevented simply by lack of spatial proximity between the relevant organisms rather than by lack of phenotypic expression by individuals.

#### (iii) MyxoEE-3

To facilitate reference to the broader evolution experiment of which the treatments examined here were a part, we refer to the overall experiment as MyxoEE-3 [7] (Myxobacteria Evolution Experiment, with '3' indicating the temporal rank of the first publication from MyxoEE-3 [69] relative to first publications from other MyxoEEs). Shared features of MyxoEE-3 treatments have been described previously [29,30,58]. Treatments examined here are summarized in the electronic supplementary material, table S1.

### (b) Strains and procedures

#### (i) Strains

In MyxoEE-3, ancestral strains differing in motility genotype and antibiotic-resistance marker were selected for increased fitness at the leading edge of expanding colonies. Multiple treatment sets of replicate populations adapted to different environmental conditions that varied in nutrient level, nutrient type or agar concentration (electronic supplementary material, table S1).

The ancestral reference strains GJV1 and GJV2 have two functional motility systems: adventurous (A) and social (S) motility (hereafter referred to as the A+S+ motility genotype) [70]. Deletion of one gene essential for either motility system led to strains that were defective in A-motility (A-S+; deletion of *cglB*; strains GJV3 and GJV5) or S-motility (A+S-; deletion of *pilA*; strains GJV4 and GJV6). GJV1, GJV3 and GJV4 are rifampicin-sensitive, whereas GJV2, GJV5 and GJV6 are rifampicin-resistant variants of the corresponding motility type (electronic supplementary material, table S1). Two distinct sub-clone genotypes of GJV1 (represented by GJV1.1 and GJV1.2) previously found to differ by one mutation were used to establish the rifampicin-sensitive A+S+ MyxoEE-3 populations and thus were examined here also [29]. No phenotypic differences between these clones were found in our assays (see the electronic supplementary material, statistical analysis). A single sub-clone was used for each of the other five ancestral strains since there are no known mutational differences between the ancestral sub-clones used to found the respective MyxoEE-3 populations.

For this study, we used MyxoEE-3 populations that evolved for 40 two-week cycles under high-nutrient conditions (CTT growth medium, see recipe below), low-nutrient conditions (0.1% casitone CTT) or with prey (*Escherichia coli* or *Bacillus subtilis* and CTT). Additionally, the agar concentration in each environment was either high (1.5%, 'HA') or low (0.5%, 'SA') (electronic supplementary material, table S1). During evolution, replicate populations derived from each of the six ancestors (GJV1-GJV6) grew and swarmed (to their ability) outwards on the surface of each selection environment for two weeks, after which a patch of defined size was collected from the leading edge of each colony and transferred to a fresh plate. Importantly, these populations never interacted with phage during MyxoEE-3.

The virulent myxophage Mx1 [57] is a double-stranded DNA Myoviridae, morphologically similar to coliphages T2 and T4 [48,71]. A single-source stock of Mx1 was generated by infecting a growing liquid culture of GJV1. We isolated phage particles with 10% chloroform and filtration (0.22 µm) and titered the resulting Mx1 stock using double agar overlay plaque assays on the highly susceptible *M. xanthus* strain DZ1 [72].

#### (ii) Assays of Mx1 growth on MyxoEE-3 ancestors and evolved populations in liquid culture

All host-quality experiments were performed in liquid CTT medium (10 mM Tris pH 8.0, 8 mM $MgSO_4$, 10 g l$^{-1}$ casitone, 1 mM $KPO_4$, pH 7.6) supplemented with 0.5 mM $CaCl_2$ in case of phage infection and incubated at 32°C and 300 rpm. Prior to each experiment, bacteria were inoculated onto CTT 1.5% agar from frozen stocks and incubated at 32°C and 90% relative humidity until sufficiently grown. Colony-edge samples were transferred to 8 ml CTT liquid and incubated shaken at 300 rpm. When the bacterial cultures reached the mid-log phase, cells were centrifuged (15 min, 5000 rpm) and resuspended with CTT liquid to approximately $2 \times 10^8$ cells ml$^{-1}$. To initiate the host-quality assays in liquid culture, aliquots of the same Mx1 phage stock at approximately $2 \times 10^6$ particles ml$^{-1}$ were added to the density-standardized bacterial populations, resulting in a multiplicity of infection (MOI) of approximately 0.01. Phages were allowed to infect both ancestral and evolved MyxoEE-3 bacterial populations for 24 h and viable phage population size after 24 h in liquid was our measure of host quality. We performed four replicates of this experiment, each divided into three randomized blocks.

To quantify and compare Mx1 population sizes at the end of the host-quality assay in liquid, we performed plaque assays on the indicator strain DZ1. DZ1 is highly susceptible to Mx1 [72] and thus maximizes the detection of viable phage particles. To isolate the phage from the bacteria, 100 µl of chloroform was added to 1 ml of liquid culture and this mixture was incubated for 5 min under constant shaking/vortexing to disrupt bacterial cells. Subsequently, dead cells and phage were separated by centrifugation (3 min, 12000 rpm). The supernatant containing the phage was stored at 4°C. Phage numbers were then estimated by serial dilution followed by mixing 10 µl of each dilution with 10 µl of the indicator strain DZ1 (approx. $10^{10}$ cells ml$^{-1}$) and 1 ml CTT 0.5% agar. This mixture was poured onto 5 ml CTT 1.5% hard agar and plates were then closed immediately and incubated until Mx1 plaques within the soft-agar-embedded lawns of DZ1 became visible and could be counted.

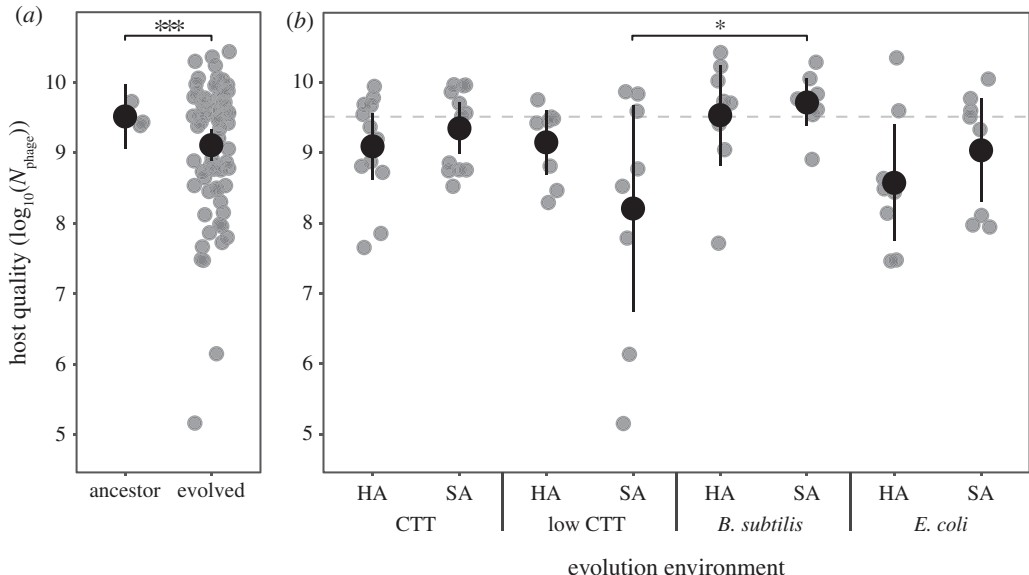

**Figure 1.** Diversification of latent host quality during phage-blind evolution occurred predominantly within rather than between selective environments. Cross-replicate means of host-quality measurements (grey circles) for ancestors and all evolved populations (*a*) and evolved populations categorized by evolution environment (*b*) with corresponding overall category means (black circles) and 95% confidence intervals. HA and SA indicate hard and soft agar environments, respectively. Host quality is measured as Mx1 phage population size 24 h after initial infection of bacterial populations (log-transformed data). The dashed line corresponds to average phage population size after growth on the experimental ancestors GJV1 and GJV2. The asterisks indicate statistically significant differences: two-sample two-sided *t*-test ((*a*) ***$p < 0.001$) and the one pairwise comparison in which treatment means differ significantly ((*b*) *post hoc* Tukey test, mixed linear model, *$p < 0.05$).

### (iii) *Myxococcus xanthus* growth in the presence of phage

To assess effects of phage on the growth of populations P65-P72 (electronic supplementary material, table S1), we measured optical density (OD 595 nm) of liquid cultures growing with and without phage. Overnight cultures of each ancestor and evolved population were diluted in equal volumes into two 15 ml cultures in 100 ml flasks, one of which was infected with phage (MOI of 0.01) and incubated shaken as described above. Measurements were taken at 0, 14, 16, 18, 20, 22 and 24 h.

### (iv) Statistical analysis

Statistical analysis is detailed in the electronic supplementary material.

## 3. Results

### (a) Phage-blind evolution lowered host quality overall while environment mildly shaped treatment means

Host quality depends on a combination of host features, including extracellular components phage must bypass or penetrate to reach the cell surface [73], surface and membrane components that phage use for invasion [49], harmful intracellular components phage must avoid or neutralize, [74] and beneficial intracellular components that phage exploit for growth. Any of these components might be altered during adaptation in the absence of phage. If selective pressures imposed by distinct MyxoEE-3 environments often differ in their effects on traits important to phage invasion and growth, MyxoEE-3 treatments might often vary in host quality. If not, more variation in host quality should be found among replicate populations within treatments than between treatment means.

Evolved populations and their ancestors were exposed to phage epidemics in shaken liquid culture for 24 h before

phage population sizes were determined to quantify host quality. On average across all 72 populations descended from the A+S+ ancestor, evolved populations supported less phage growth than their ancestors (figure 1*a*; electronic supplementary material, figure S1; $t_{54} = -4.78$, $p < 0.001$). The observed trend of decreased host quality suggests that adaptation to laboratory conditions generally increased resistance to a major natural stress (in this case phage). This is an intriguing outcome, as laboratory 'domestication' of natural isolates usually relaxes selection for natural stresses [26,75], often resulting in corresponding trait losses [9,76,77]. Nonetheless, evolved bacterial populations might have become lower quality hosts by two very different mechanisms—either by becoming more rapidly killed by phage and thereby supporting less phage growth overall or by individual bacteria becoming more resistant to phage. We investigate these hypotheses for one MyxoEE-3 treatment below. However, because absolute fitness is what matters from the phage perspective, our primary emphasis is on host quality *per se*, irrespective of what specific traits underlie its unselected evolution.

Despite the overall decrease in host quality across all evolved populations, no individual treatment considered separately changed significantly from the ancestor (Dunnett contrasts, all $p > 0.1$), reflecting variable outcomes among replicate populations within treatments (figure 1). However, the selection environment did nonetheless have a small but significant effect on the structure of evolved host quality outcomes (figure 1*b*; mixed linear model, $F_{7,64} = 2.4$, $p = 0.03$). This effect was driven predominantly by a difference between two environments—populations that evolved with *B. subtilis* as prey on SA were higher quality hosts, on average, than populations evolved on low-nutrient SA (figure 1*b*; *post hoc* multiple comparisons with Tukey method for *p*-value adjustment $t_{66} = -3.24$, $p = 0.04$).

We further tested whether environmental features shared across subsets of treatments affected average host-quality

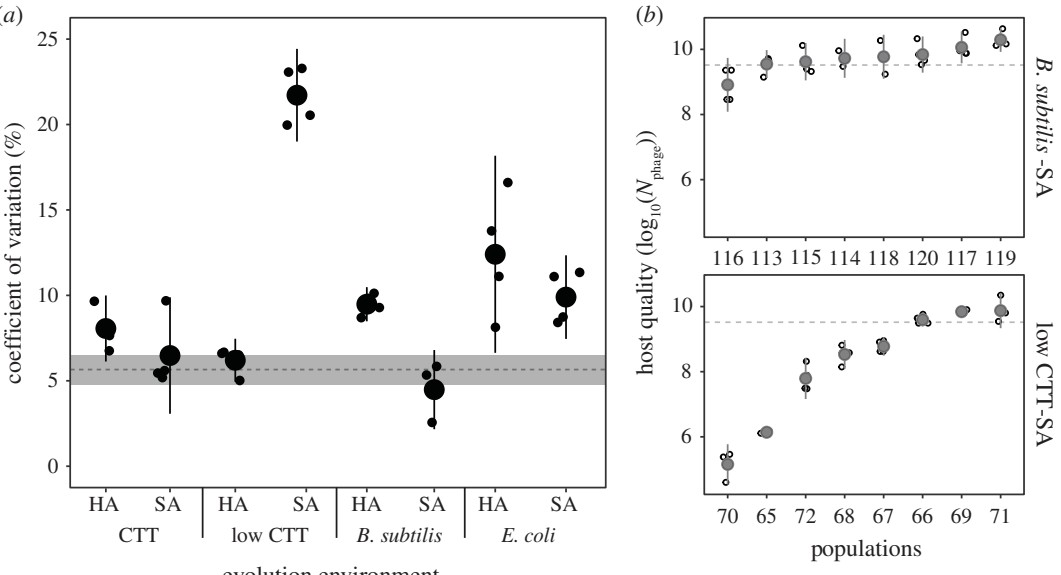

**Figure 2.** Selective environments differentially constrained stochastic diversification of host quality within treatments. (*a*) Within-treatment coefficients of variation (CVs) of phage population size 24 h post infection. Small and large circles represent within-replicate-assay CV estimates across evolved populations and cross-replicate-assay means for each treatment, respectively. HA and SA differentiate hard and soft agar environments, respectively. Error bars show 95% confidence intervals. For comparison, the dashed line indicates the between-treatment CV (i.e. the cross-replicate-assay average of the CV among host-quality means for each treatment). Grey shaded area is the corresponding 95% confidence interval. (*b*) Host quality of evolved populations from the least (upper panel) and most (lower panel) evolutionarily diversified treatments. Grey circles are the means across four biological replicates (open circles) and error bars represent 95% confidence intervals.

evolution and thus grouped treatments by nutrient type (high- and low-casitone CTT, *B. subtilis* or *E. coli*) and agar type (HA or SA), irrespective of other factors. Agar type did not influence mean host-quality evolution (electronic supplementary material, figure S2*a*; $F_{1,71} = 0.028$, $p = 0.87$) but nutrient type did (electronic supplementary material, figure S2*b*; $F_{3,69} = 3.73$, $p = 0.015$). The latter outcome is caused predominantly by the low-nutrient and *B. subtilis* subsets, with populations having grown at low-nutrient levels evolving lower host quality than populations that evolved with *B. subtilis* as prey (post hoc multiple comparisons adjusted with the Tukey method, $t_{71} = 2.9$, $p = 0.02$).

## (b) Degrees of within-treatment diversification varied greatly across selective environments

Although divergence among treatment means due to environmental differences was limited, we noted a high degree of diversification among populations overall that was not evenly distributed across treatments. To visualize host-quality diversification at multiple levels, we compared the coefficient of variation within versus between selective environments for all eight treatments with A+S+ ancestors. Variation among populations within environments (on average 9.9%, ranging from 4.5 to 21.8%; figure 2*a*) greatly exceeded variation across environments (5.7%, calculated among within-treatment means, $t_{38} = 4.11$, $p < 0.001$), further confirming that chance differences in the mutational trajectories of replicate populations contributed more to overall diversification than did systematic differences between treatments.

Environments might differ in the degree to which they allow latent phenotypes to diverge stochastically among replicate populations if (i) LPE is caused largely by mutations that evolved owing to selection rather than drift and (ii) the adaptive landscapes of distinct environments differ in their ranges of accessible adaptive pathways with regard to their

indirect effects on host quality. We found that the degree of host-quality diversification among replicate populations adapted to the same environment varied greatly across treatments (figure 2*a*; electronic supplementary material, figure S1; $F_{7,21} = 48.83$, $p < 0.001$). At one extreme, populations evolved on SA with *B. subtilis* diverged very little in host quality (figure 2). At the other extreme, populations evolved on low-nutrient SA diversified much more than populations in any other treatment (figure 2). Such variation in diversification across environments indicates that many of the mutations underlying LPE evolved because of selection and that distinct adaptive landscapes differ in how much they constrain latent-phenotype divergence.

## (c) Some populations latently evolved nearly complete resistance to phage antagonism

Post-infection phage population sizes varied by more than 10-fold across replicate evolved host populations within all treatments, more than 100-fold in five treatments and nearly five orders of magnitude in one treatment (low-nutrient SA, figure 1; electronic supplementary material, figure S1). Given such diversity, we tested whether bacterial growth would be suppressed to a degree inversely correlated with phage population growth. It is not obvious that such a correlation will occur, because, as noted above, phage growth might be low on both highly susceptible and highly resistant bacterial genotypes and thereby prevent a correlation. On highly susceptible hosts, phage growth may be low because phage strongly suppress the increase of their only growth substrate. By contrast, phage will not grow much from even large populations of highly resistant hosts. In this scenario, phage productivity could be maximal on bacterial populations exhibiting intermediate growth in the presence of phage.

During experimental epidemics with evolved populations from the most diversified evolutionary treatment (P65–P72,

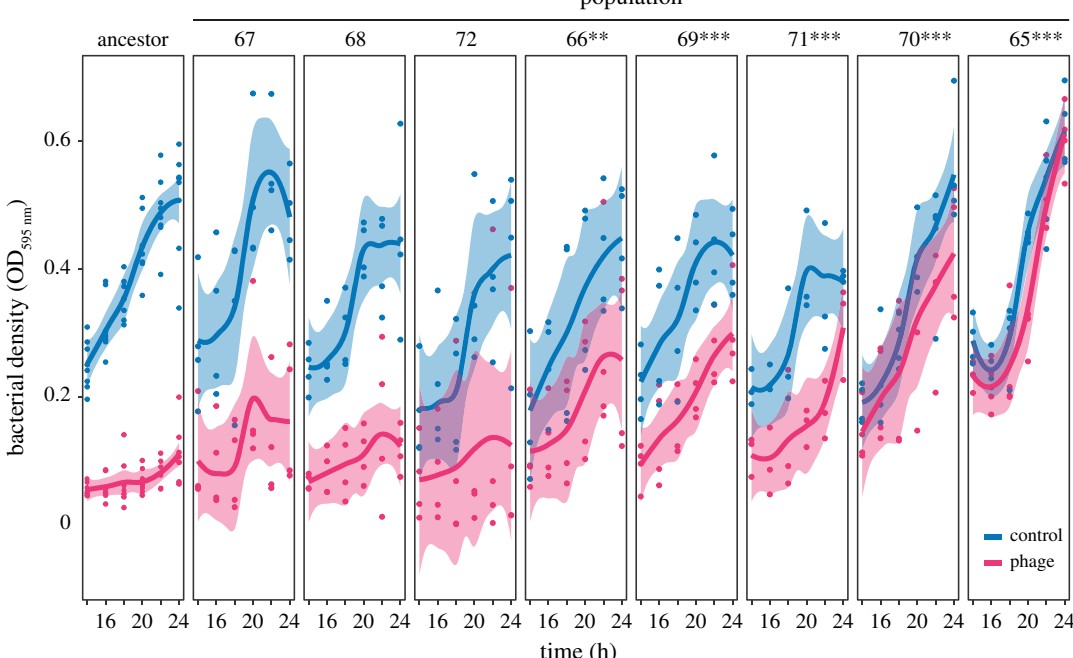

**Figure 3.** Diversity of evolved bacterial resistance to growth suppression by phage. Growth of the ancestors and of MyxoEE-3 populations evolved on low-nutrient soft agar (P65-P72) in the presence (red) and absence (blue) of phage. Data points show optical density ($OD_{595\,nm}$) measurements over time for four temporally separate biological replicates, trendlines track conditional mean values of locally weighted regressions and shaded areas represent 95% confidence intervals of the fit. The asterisks indicate significant differences to the ancestors (Dunnett test, mixed linear model, **$p < 0.01$ and ***$p < 0.001$).

low-nutrient SA), we tracked both phage growth and *M. xanthus* population dynamics, and the latter in comparison to bacterial growth in the absence of phage. None of these evolved populations grew less in the presence of phage than their ancestor, indicating that increased susceptibility to phage killing was not a general mechanism by which host quality often decreased during MyxoEE-3 (figure 3). As expected, these populations varied greatly in the degree to which Mx1 hindered their growth relative to their growth in the absence of phage (figure 3). Three evolved populations (P67, P68 and P72), like the ancestors, grew very little over the 24 h epidemics, both relative to the phage-free controls and in absolute numbers. The other five populations all grew significantly more than their ancestors, again both relative to the phage-free controls and in absolute numbers (Dunnett test against the ancestor, all *p*-values < 0.005). The two evolved populations that supported the least phage growth (P65 and P70, figure 2*b*) exhibited the highest bacterial growth, which was not significantly lower than phage-free growth after 24 h. Thus, complete (or nearly complete) resistance to viral load is found to evolve indirectly.

In contrast with the hypothetical scenario presented above, total phage productivity was found to weakly correlate with bacterial growth reduction (Spearman's rho correlation $r_S = 0.31$, $n = 40$, $p = 0.058$; electronic supplementary material, figure S3). However, large numbers of phage were able to grow from bacterial populations that exhibited very different degrees of growth suppression by phage. For example, Mx1 consistently grew to large population sizes on the ancestors, P66-P69 and P71, yet these populations varied greatly in the degree to which their growth was suppressed by phage. These results reveal idiosyncrasy in relationships between host growth and phage growth and thus point to those relationships evolving by diverse molecular mechanisms.

Intriguingly, we also noted substantial evolution and diversification of growth dynamics among the evolved populations in this treatment in the absence of phage, with several (e.g. P67, P68, and P71) slowing or ceasing growth earlier than the ancestors and other evolved populations (e.g. P65 and P70). Thus, variation of mutational input across replicate populations generated substantial diversification of growth dynamics in liquid media in the absence of phage.

### (d) Ancestral motility genotype determines ancestral host quality, mean host-quality evolution and within-treatment host-quality diversification

Motility not only allows organisms to search for new resources but also allows active flight from biotic and abiotic danger. In bacteria, motility can allow cells to escape from non-motile phage particles [78]. On the other hand, motility-related surface structures such as type-IV pili can also make bacteria susceptible to phage attack by acting as phage receptors [79]. To our knowledge, the relationship between motility and host–phage interactions, either behaviourally or evolutionarily, has yet to be examined for bacteria with multiple motility systems. We exploited the design of MyxoEE-3, which included an ancestor with both *M. xanthus* motility systems intact (A+S+) and also ancestors lacking a gene essential for either motility system (A-S+, Δ*cglB* and A+S-, Δ*pilA*, electronic supplementary material, table S1), to test for motility-genotype effects on ancestral host quality and subsequent host-quality evolution.

We quantified total phage productivity after 24 h of growth in liquid on all motility-genotype ancestors and all descendant populations that evolved on CTT HA or SA during MyxoEE-3. The absence of *cglB* in the A-S+ ancestors had no effect on phage growth (motility effect in mixed linear

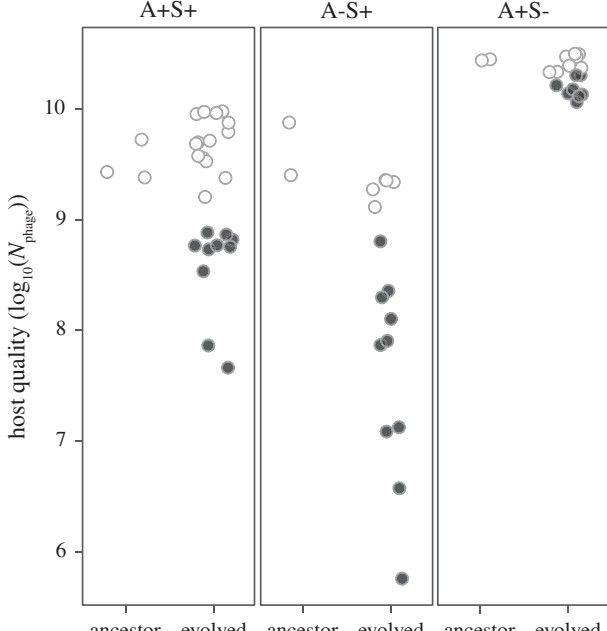

**Figure 4.** Ancestral motility genotype determined both degree of trait mean evolution and degree of stochastic diversification for host quality. Phage population size 24 h after infection of ancestors and evolved populations with both motility systems intact (A+S+) or lacking either system (A-S+ or A+S-). Each data point represents the mean of four biological replicates. Colours show the difference between evolved populations and their respective ancestors (open circles: non-significant difference, black circles: significant difference; Dunnett test, mixed linear model).

model $F_{2,4} = 12.47$, $p = 0.019$, post hoc contrasts $t_4 = -0.66$, $p = 0.8$), whereas the absence of *pilA* in the A+S- ancestors increased phage productivity nearly 10-fold (post hoc contrasts $t_4 = 4.79$, $p = 0.019$; figure 4). Thus, the production of pilin, but not of CglB, greatly reduces a phage epidemic.

Ancestral motility genotype also affected the character of host-quality evolution. Mean host quality of A-S+ populations decreased significantly from their ancestral values during evolution ($t_{55} = 2.83$, $p = 0.007$) and decreased significantly more than did the host quality of A+S+ or A+S- populations, while A+S+ populations decreased more than did A+S- populations ($F_{2,54} = 8.34$, $p < 0.001$, all post hoc contrast $p$-values $< 0.001$). Moreover, ancestral motility genotype also affected degrees of diversification within treatments, with evolved host quality spanning over four orders of magnitude among the A-S+ populations but less than a factor of ten among the A+S- populations (figure 4). The large differences in both ancestral phenotypes and evolutionary patterns between the A+S- category of populations versus the two categories with intact S-motility indicate that pilin production potentiates latent evolutionary reduction and diversification of host quality.

## 4. Discussion

Previously, Meyer *et al.* found that the *Escherichia coli* long-term evolution experiment (LTEE) populations adapting to one selective environment in the absence of phage evolved changes in their interactions with multiple phage types [26]. Replicate LTEE populations evolved increased susceptibility towards phage T6* and increased resistance towards bacteriophage lambda with some degree of parallelism. Here, we

asked whether multiple distinct phage-free selective environments might differentially shape how bacterial populations descended from a common ancestor would diverge from each other in quality as phage hosts. We tested for such inter-treatment divergence with respect to both mean host quality and the degree of stochastic within-treatment diversification among replicate populations.

Overall, MyxoEE-3 populations tended to support less phage growth than their ancestors and diversified greatly, including some lineages that evolved nearly complete resistance to negative effects of phage on bacterial population growth. We found high degrees of diversification in host quality among replicate populations within treatments. Indeed, among A+S+ populations, distinct selective environments drove only a small degree of divergence between selective-environment treatments in mean host quality; the primary diversifying force was chance variation in mutational input across replicate populations (figures 1 and 2).

Yet despite the limited effect of selection on the divergence of treatment means, selection nonetheless strongly shaped the character of intrinsically non-adaptive diversification of latent phenotypes. Specifically, selective environments determined the degree to which replicate populations diversified in latent host quality, thus indicating that many of the mutations driving such divergence first evolved owing to selection. Further, this result indicates that distinct adaptive landscapes can differ not only in the number of adaptive mutational pathways replicate populations might follow [60,80,81], but concomitantly can also differ in the range of latent phenotypic effects generated by those adaptive pathways [7]. Thus, distinct natural environments may often differ in the character of latent phenotypic diversity they allow to evolve.

Determination of latent stochastic diversification of host–parasite interactions by environment-specific features of fitness landscapes might apply not only to host evolution but to parasite evolution also. For example, consider a scenario with animal viruses in which different initial host species for a given viral type differ in fitness landscape structure and thereby allow different ranges of adaptive mutational pathways to be followed by evolving viral populations. Such differences might in turn generate differences in latent virus–host interaction phenotypes, including potential for jumps to novel host species (e.g. zoonosis).

We further investigated whether ancestral motility genotype is important to ancestral host–parasite interactions and/or their subsequent evolution, including the extent of diversification (figure 4). While type-IV pili are the very means of cell invasion by phage in some host species [49,50], we found that the production of pilin, the building-block of type-IV pili [51], both greatly reduces phage population growth in our ancestral genetic background and promotes greater latent evolutionary reduction and diversification of host quality than occurs in populations lacking pilin (figure 4). These immediate and evolutionary effects, respectively, of pilin production may be mediated not by pilin *per se* but by the *M. xanthus* exopolysaccharide (EPS) matrix, which is positively regulated by pilin production [69,82]. The EPS matrix is necessary for effective S-motility [83] and mediates cell–cell adhesion [84]. We hypothesize that the EPS matrix hinders Mx1 access to its adsorption receptor (which remains unknown), thereby explaining the nearly 10-fold increase in phage growth resulting from deletion of *pilA* to generate the A+S- genotype. This hypothesis would suggest that in

A+S+ and A-S+ ancestors, evolution of EPS matrix may have often provided greater protection against phage compared to ancestral EPS, thereby promoting greater diversification than among populations lacking ancestral EPS.

It has long been recognized that forces other than direct selection on focal traits play important roles in shaping evolutionary diversification [5,13,23,85], but latently evolved diversification is only rarely quantified [7,15,26]. We have shown that host–parasite interactions diversify greatly during parasite-blind evolution, highlighting the need to more deeply integrate LPE into our overall conception of biological diversification. The total long-term phenotypic diversification of populations evolved in an original focal context can be conceived to include both the divergence already actualized phenotypically in that original context and the sum of all latent phenotypic diversification revealed only later in new contexts. This expansive view of diversification can, in turn, inform how conservation efforts are conceived [86] to include conservation of latent phenotypes and corresponding evolutionary potential.

Data accessibility. Raw data and representative code are available from the Dryad Digital Repository: https://doi.org/10.5061/dryad. rn8pk0p86 [87].

Authors' contributions. L.F. designed experiments, carried out experiments and co-wrote the manuscript. M.V. designed experiments, performed the statistical analysis and co-wrote the manuscript. G.J.V. designed experiments and co-wrote the manuscript.

Competing interest. The authors declare no conflict of interest.

Funding. This work was funded in part by Swiss National Science Foundation (SNSF) grants 31003A/B_16005 to G.J.V. and an ETH Fellowship 16-2 FEL-59 to M.V.

Acknowledgements. The authors thank Peter Ashcroft, Marco La Fortezza, Samay Pande, Joshua Payne, Sébastien Wielgoss and all members of the ETH Zürich Evolutionary Biology group for helpful discussions and support.

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
