## [Peer Review File · Proceedings of the Royal Society B: Biological Sciences]

Review History

RSPB-2020-2773.R0 (Original submission)

Review form: Reviewer 1

Recommendation

Major revision is needed (please make suggestions in comments)

Scientific importance: Is the manuscript an original and important contribution to its field?

Good

General interest: Is the paper of sufficient general interest?

Good

Quality of the paper: Is the overall quality of the paper suitable?

Acceptable

Is the length of the paper justified?

Yes

Should the paper be seen by a specialist statistical reviewer?

No

Do you have any concerns about statistical analyses in this paper? If so, please specify them explicitly in your report.

Yes

It is a condition of publication that authors make their supporting data, code and materials available - either as supplementary material or hosted in an external repository. Please rate, if applicable, the supporting data on the following criteria.

Is it accessible?

N/A

Is it clear?

N/A

Is it adequate?

N/A

Do you have any ethical concerns with this paper?

No

Comments to the Author

The authors use experimental evolution of a social bacterium (*Myxococcus xanthus*) to analyse the allopatric diversification of latent host quality. They found that evolving the bacterium in the absence of phage reduced the quality of the host, from the perspective of the phage. In addition, the authors highlight that latent-phenotype evolution, a term they coin, should be used to inform host-parasite evolutionary theory.

- Need more information on statistical analysis in the main body of the paper (line 267-268).
- Figures 1-4: The figures are extremely small.

Supplementary material - Statistical analysis:

- I think it would be better for the statistical analysis to be included in the main manuscript.
- Where the authors fit random effects in their models - they should clearly state whether these were random effects on the intercept or random effects on the gradient.
- In the second paragraph of statistical analysis the authors don't explicitly state what the response variable is.
- In the second paragraph of the statistical analysis, the authors fit two fixed effects, but did they fit an interaction between these effects? And if not, why did the authors choose to not fit an interaction?

Minor comments:

- Line 22: there is a 'II' but no 'I'
- Line 94: 'has been little investigated' - I understand what the authors mean here but the phrasing is a little odd.
- Line 273-285 - Perhaps this paragraph would be better suited for the discussion for discussion as there are no results here. Results don't begin until line 288.
- Line 295-309 and Line 430-444 - Again these sections are more discussion based than results focussed.
- Line 471 - write out "long term evolution experiment" before the acronym.
- Fig 1B and 2A - include "HA" and "SA" in figure legend - I assume these refer to hard and soft agar, but it would be good to include this in the figure legend for clarity.
- Supplementary info: Final line of the statistical analysis - the references are not provided for all of the R packages (ggpubr and ggsignif).

Review form: Reviewer 2

Recommendation

Major revision is needed (please make suggestions in comments)

Scientific importance: Is the manuscript an original and important contribution to its field?

Good

General interest: Is the paper of sufficient general interest?

Good

Quality of the paper: Is the overall quality of the paper suitable?

Good

Is the length of the paper justified?

Yes

Should the paper be seen by a specialist statistical reviewer?

No

Do you have any concerns about statistical analyses in this paper? If so, please specify them explicitly in your report.

No

It is a condition of publication that authors make their supporting data, code and materials available - either as supplementary material or hosted in an external repository. Please rate, if applicable, the supporting data on the following criteria.

Is it accessible?

N/A

Is it clear?

N/A

Is it adequate?

N/A

Do you have any ethical concerns with this paper?

No

Comments to the Author

Dear Authors,

I think that this is a paper of interest and is generally well written. I enjoyed reading it and I like how the latent phenotype and LPE were discussed and defined.

However, I have struggled to understand the experimental set-up and rationale for using the DZ strain for quantifying phages after the 24 hour "host quality experiment"....that is if I have interpreted the experimental set up correctly. Please see my comments on this below. Until I could get some clarity on the experimental setup and the rationale behind it is hard to properly evaluate the results and discussion.

In line 90-91... you define host quality as the degree to which a host genotype or population facilitates parasite growth. Am I correct in thinking that for phage quantification following your 24 hour phage propagation/host quality experiment that you used strain DZ1 as the only host to

quantify the phages?

If this is the case then I think that there may be a fundamental problem with this approach due to efficiency of plaquing (EOP) which can vary widely between bacterial hosts. With EOP you can get huge variance in phage titres (by many orders of magnitude) for a single phage stock when tested on highly related hosts.

For this reason you need to quantify phage on the host that you are interested in assessing for host quality not on a "common garden" host e.g. DZ2. By using DZ2 for quantification of the phage populations size on a given host (as a proxy for host quality) you do not have a measure of host quality on that host, you only have a measure of how it titres on DZ2 - i.e. you could have some phages that grow really well on their own host but not on DZ2 and vice versa. I hope this point makes sense as it is key to the experiment but please correct me if I have misinterpreted the design.

In addition to this, another key issue for me is the use of the 24 hour phage propagation experiment as a measure of host quality. 24 hours is a long time from a bacterial and phage perspective and you could get selection for resistance amongst other things during this time frame. I am not sure that it is a good and accurate measure of host quality as you define it. Perhaps a one more useful measure(s) of host quality would be to take the hosts you are interested in and doing a one-step growth curve with them... I think this would provide a much better measure of host quality as it you will get information on phage adsorption, length of infection cycle and burst size which will allow you measure the number of progeny produced per host and over what time-scale. I think this would really bolster your study as it would remove any ambiguity or alternative explanation as to why you see lower populations on different evolved hosts.

Another issue was with the initial experimental set-up that I am not sure I understood properly - were six different ancestral variants used to initiate the experiment? Line 221

Also, Line 104 onwards - you discuss the potential receptors sites of phages but do you have any detailed information on how the phage used in this study infects *M. xanthus* and how resistance evolves - this might help inform the data that you have observed for certain selective environments? For example you investigate the *M. xanthus* motility systems - is this because phage bind to flagella or pilli or what was the motivation for using these variants in the analysis.

Decision letter (RSPB-2020-2773.R0)

06-Feb-2021

Dear Ms Freund:

I am writing to inform you that your manuscript RSPB-2020-2773 entitled "Hidden paths to endless forms most wonderful: Parasite-blind diversification of host quality" has, in its current form, been rejected for publication in Proceedings B. Both reviewers and the AE have provided comments on your manuscript (which you can find below), and while they, and I, like your premise, there are a number of issues that need to be clarified before we can further consider your manuscript. With this in mind we would be happy to consider a resubmission, provided the comments of the referees are fully addressed. However please note that this is not a provisional acceptance and we will send the revision back to the original reviewers, should the be available.

The resubmission will be treated as a new manuscript. However, we will approach the same reviewers if they are available and it is deemed appropriate to do so by the Editor. Please note

that resubmissions must be submitted within six months of the date of this email. In exceptional circumstances, extensions may be possible if agreed with the Editorial Office. Manuscripts submitted after this date will be automatically rejected.

Sincerely,
 Dr Sarah Brosnan
 Editor, Proceedings B
 mailto: proceedingsb@royalsociety.org

Associate Editor
 Board Member: 1
 Comments to Author:

This paper presents interesting data on how latent phenotypes that evolve (either neutrally or non-neutrally) in non-focal environments are then expressed and thus shape interspecific outcomes in novel environments where the interactor is present, delving into an interesting question in evolution and ecology. Both reviewers found the topic addressed and the findings interesting. Reviewer 1 had several important questions about the statistical analyses and ways to clarify what was done and why, while Reviewer 2 had several fundamental questions on how the work was performed and the implications of these choices, all of which should be addressed. I echo the comments of Reviewer 1 that the paper will need some reworking for clarification and to have the impact required for publication in Proc B.

Reviewer(s)' Comments to Author:

Referee: 1

Comments to the Author(s)

The authors use experimental evolution of a social bacterium (*Myxococcus xanthus*) to analyse the allopatric diversification of latent host quality. They found that evolving the bacterium in the absence of phage reduced the quality of the host, from the perspective of the phage. In addition, the authors highlight that latent-phenotype evolution, a term they coin, should be used to inform host-parasite evolutionary theory.

- Need more information on statistical analysis in the main body of the paper (line 267-268).
- Figures 1-4: The figures are extremely small.

Supplementary material - Statistical analysis:

- I think it would be better for the statistical analysis to be included in the main manuscript.

- Where the authors fit random effects in their models - they should clearly state whether these were random effects on the intercept or random effects on the gradient.
- In the second paragraph of statistical analysis the authors don't explicitly state what the response variable is.
- In the second paragraph of the statistical analysis, the authors fit two fixed effects, but did they fit an interaction between these effects? And if not, why did the authors choose to not fit an interaction?

Minor comments:

- Line 22: there is a 'II' but no 'I'
- Line 94: 'has been little investigated' - I understand what the authors mean here but the phrasing is a little odd.
- Line 273-285 - Perhaps this paragraph would be better suited for the discussion for discussion as there are no results here. Results don't begin until line 288.
- Line 295-309 and Line 430-444 - Again these sections are more discussion based than results focussed.
- Line 471 - write out "long term evolution experiment" before the acronym.
- Fig 1B and 2A - include "HA" and "SA" in figure legend - I assume these refer to hard and soft agar, but it would be good to include this in the figure legend for clarity.
- Supplementary info: Final line of the statistical analysis - the references are not provided for all of the r packages (ggpubr and ggsignif).

Referee: 2

Comments to the Author(s)

Dear Authors,

I think that this is a paper of interest and is generally well written. I enjoyed reading it and I like how the latent phenotype and LPE were discussed and defined.

However, I have struggled to understand the experimental set-up and rationale for using the DZ strain for quantifying phages after the 24 hour "host quality experiment"....that is if I have interpreted the experimental set up correctly. Please see my comments on this below. Until I could get some clarity on the experimental setup and the rationale behind it is hard to properly evaluate the results and discussion.

In line 90-91... you define host quality as the degree to which a host genotype or population facilitates parasite growth. Am I correct in thinking that for phage quantification following your 24 hour phage propagation/host quality experiment that you used strain DZ1 as the only host to quantify the phages?

If this is the case then I think that there may be a fundamental problem with this approach due to efficiency of plaquing (EOP) which can vary widely between bacterial hosts. With EOP you can get huge variance in phage titres (by many orders of magnitude) for a single phage stock when tested on highly related hosts.

For this reason you need to quantify phage on the host that you are interested in assessing for host quality not on a "common garden" host e.g. DZ2. By using DZ2 for quantification of the phage populations size on a given host (as a proxy for host quality) you do not have a measure of host quality on that host, you only have a measure of how it titres on DZ2 - i.e. you could have some phages that grow really well on their own host but not on DZ2 and vice versa. I hope this point makes sense as it is key to the experiment but please correct me if I have misinterpreted the design.

In addition to this, another key issue for me is the use of the 24 hour phage propagation experiment as a measure of host quality. 24 hours is a long time from a bacterial and phage

perspective and you could get selection for resistance amongst other things during this time frame. I am not sure that it is a good and accurate measure of host quality as you define it.

Perhaps a one more useful measure(s) of host quality would be to take the hosts you are interested in and doing a one-step growth curve with them... I think this would provide a much better measure of host quality as it you will get information on phage adsorption, length of infection cycle and burst size which will allow you measure the number of progeny produced per host and over what time-scale. I think this would really bolster your study as it would remove any ambiguity or alternative explanation as to why you see lower populations on different evolved hosts.

Another issue was with the initial experimental set-up that I am not sure I understood properly - were six different ancestral variants used to initiate the experiment? Line 221

Also, Line 104 onwards - you discuss the potential receptors sites of phages but do you have any detailed information on how the phage used in this study infects *M. xanthus* and how resistance evolves - this might help inform the data that you have observed for certain selective environments? For example you investigate the *M. xanthus* motility systems - is this because phage bind to flagella or pilli or what was the motivation for using these variants in the analysis.

Author's Response to Decision Letter for (RSPB-2020-2773.R0)

See Appendix A.

RSPB-2021-0456.R0

Review form: Reviewer 1

Recommendation

Accept as is

Scientific importance: Is the manuscript an original and important contribution to its field?

Good

General interest: Is the paper of sufficient general interest?

Good

Quality of the paper: Is the overall quality of the paper suitable?

Good

Is the length of the paper justified?

Yes

Should the paper be seen by a specialist statistical reviewer?

No

Do you have any concerns about statistical analyses in this paper? If so, please specify them explicitly in your report.

No

It is a condition of publication that authors make their supporting data, code and materials available - either as supplementary material or hosted in an external repository. Please rate, if applicable, the supporting data on the following criteria.

Is it accessible?

Yes

Is it clear?

Yes

Is it adequate?

Yes

Do you have any ethical concerns with this paper?

No

Comments to the Author

I am satisfied that the authors addressed my comments and have revised the manuscript appropriately.

Review form: Reviewer 2

Recommendation

Accept as is

Scientific importance: Is the manuscript an original and important contribution to its field?

Excellent

General interest: Is the paper of sufficient general interest?

Good

Quality of the paper: Is the overall quality of the paper suitable?

Excellent

Is the length of the paper justified?

Yes

Should the paper be seen by a specialist statistical reviewer?

No

Do you have any concerns about statistical analyses in this paper? If so, please specify them explicitly in your report.

No

It is a condition of publication that authors make their supporting data, code and materials available - either as supplementary material or hosted in an external repository. Please rate, if applicable, the supporting data on the following criteria.

Is it accessible?

Yes

Is it clear?

Yes

Is it adequate?

Yes

Do you have any ethical concerns with this paper?

No

Comments to the Author

Thank you for your detailed clarification, it has greatly helped with the interpretation of the results and conclusions reached. within the confines of your definition of host quality I think the experiments are perfectly fine but perhaps in future studies some one step growth curves could provide more detailed insight.

Decision letter (RSPB-2021-0456.R0)

17-Mar-2021

Dear Dr Freund

I am pleased to inform you that your Review manuscript RSPB-2021-0456 entitled "Hidden paths to endless forms most wonderful: Parasite-blind diversification of host quality" has been accepted for publication in Proceedings B.

The referee(s) do not recommend any further changes. Therefore, please proof-read your manuscript carefully and upload your final files for publication. Because the schedule for publication is very tight, it is a condition of publication that you submit the revised version of your manuscript within 7 days. If you do not think you will be able to meet this date please let me know immediately.

To upload your manuscript, log into <http://mc.manuscriptcentral.com/prsb> and enter your Author Centre, where you will find your manuscript title listed under "Manuscripts with Decisions." Under "Actions," click on "Create a Revision." Your manuscript number has been appended to denote a revision.

You will be unable to make your revisions on the originally submitted version of the manuscript. Instead, upload a new version through your Author Centre.

- 1) A text file of the manuscript (doc, txt, rtf or tex), including the references, tables (including captions) and figure captions. Please remove any tracked changes from the text before submission. PDF files are not an accepted format for the "Main Document".
- 2) A separate electronic file of each figure (tiff, EPS or print-quality PDF preferred). The format should be produced directly from original creation package, or original software format. Please note that PowerPoint files are not accepted.
- 3) Electronic supplementary material: this should be contained in a separate file from the main text and the file name should contain the author's name and journal name, e.g. `authorname_procb_ESM_figures.pdf`

All supplementary materials accompanying an accepted article will be treated as in their final form. They will be published alongside the paper on the journal website and posted on the online figshare repository. Files on figshare will be made available approximately one week before the

accompanying article so that the supplementary material can be attributed a unique DOI. Please see: <https://royalsociety.org/journals/authors/author-guidelines/>

4) Data-Sharing and data citation

It is a condition of publication that data supporting your paper are made available. Data should be made available either in the electronic supplementary material or through an appropriate repository. Details of how to access data should be included in your paper. Please see <https://royalsociety.org/journals/ethics-policies/data-sharing-mining/> for more details.

<http://datadryad.org/submit?journalID=RSPB&manu=RSPB-2021-0456> which will take you to your unique entry in the Dryad repository.

Once again, thank you for submitting your manuscript to Proceedings B and I look forward to receiving your final version. If you have any questions at all, please do not hesitate to get in touch.

Sincerely,
Dr Sarah Brosnan
Editor, Proceedings B
<mailto:proceedingsb@royalsociety.org>

Reviewer(s)' Comments to Author:

Referee: 1

Comments to the Author(s).

I am satisfied that the authors addressed my comments and have revised the manuscript appropriately.

Referee: 2

Comments to the Author(s).

Thank you for your detailed clarification, it has greatly helped with the interpretation of the results and conclusions reached. within the confines of your definition of host quality I think the experiments are perfectly fine but perhaps in future studies some one step growth curves could provide more detailed insight.

Decision letter (RSPB-2021-0456.R1)

24-Mar-2021

Dear Ms Freund

I am pleased to inform you that your manuscript entitled "Hidden paths to endless forms most wonderful: Parasite-blind diversification of host quality" has been accepted for publication in Proceedings B.

Data Accessibility section

Open Access

Paper charges

Sincerely,

Proceedings B

Appendix A

RSPB-2020-2733

Hidden paths to endless forms most wonderful: Parasite-blind diversification of host quality

Lisa Freund, Marie Vasse and Gregory J. Velicer,

Response to Referees

Associate Editor

Board Member: 1

Comments to Author:

This paper presents interesting data on how latent phenotypes that evolve (either neutrally or non-neutrally) in non-focal environments are then expressed and thus shape interspecific outcomes in novel environments where the interactor is present, delving into an interesting question in evolution and ecology. Both reviewers found the topic addressed and the findings interesting. Reviewer 1 had several important questions about the statistical analyses and ways to clarify what was done and why, while Reviewer 2 had several fundamental questions on how the work was performed and the implications of these choices, all of which should be addressed. I echo the comments of Reviewer 1 that the paper will need some reworking for clarification and to have the impact required for publication in Proc B.

We thank the editor and both referees for their thoughtful and positive comments. We respond to the referee comments below and have made corresponding manuscript changes in most cases. In particular, we have sought to make our statistical analyses clearer in response to Referee 1 and, in response to Referee 2, have sought to better clarify our experimental procedures and the importance of using a single highly susceptible strain of *M. xanthus* (DZ1) to quantify populations sizes of Mx1 phage after 24 h of growth in liquid cultures of various MyxoEE-3 populations, which is our measure of host quality in the context of this study.

Referee: 1

Comments to the Author(s)

*The authors use experimental evolution of a social bacterium (*Myxococcus xanthus*) to analyse the allopatric diversification of latent host quality. They found that evolving the bacterium in the absence of phage reduced the quality of the host, from the perspective of the phage. In addition, the authors highlight that latent-phenotype evolution, a term they coin, should be used to inform host-parasite evolutionary theory.*

- Need more information on statistical analysis in the main body of the paper (line 267-268).

We would have liked to move the statistical material to the main text, but we consulted the editor on this point and were told that the restriction on the page number is very strict, and therefore we regretfully need to keep the statistics in the supplement.

- Figures 1-4: The figures are extremely small.

We noticed the same after submission and regret this. The figures were regular size when we uploaded them and were somehow made smaller by the Proc B file conversion process. We will discuss this issue with the production editor to make sure the figures are adequately sized for publication if the paper is accepted.

Supplementary material - Statistical analysis:

- I think it would be better for the statistical analysis to be included in the main manuscript.

See reply above.

- Where the authors fit random effects in their models - they should clearly state whether these were random effects on the intercept or random effects on the gradient.

We fitted on the intercept and now indicate this in the manuscript.

- In the second paragraph of statistical analysis the authors don't explicitly state what the response variable is.

We now indicate that the response variable is the final number of phage particles after 24 h of growth in liquid cultures of various *M. xanthus* populations (log-transformed data).

- In the second paragraph of the statistical analysis, the authors fit two fixed effects, but did they fit an interaction between these effects? And if not, why did the authors choose to not fit an interaction?

We did fit the interaction between the two fixed effects and now clearly indicate this in the text.

Minor comments:

- Line 22: there is a 'II)' but no 'I)'

This has been corrected.

- Line 94: 'has been little investigated' – I understand what the authors mean here but the phrasing is a little odd.

We prefer this phrasing over alternatives.

- Line 273-285 – Perhaps this paragraph would be better suited for the discussion for discussion as there are no results here. Results don't begin until line 288.

- Line 295-309 and Line 430-444 – Again these sections are more discussion based than results focussed.

We think keeping this material in the Results section is helpful for assisting readers with interpreting the results as we describe them.

- Line 471 - write out "long term evolution experiment" before the acronym.

We have written this out.

- Fig 1B and 2A - include "HA" and "SA" in figure legend – I assume these refer to hard and soft agar, but it would be good to include this in the figure legend for clarity.

We now explain these abbreviations in the figure legend.

- Supplementary info: Final line of the statistical analysis – the references are not provided for all of the *r* packages (*ggpubr* and *ggsignif*).

We added the missing references.

Referee: 2

Comments to the Author(s)

Dear Authors,

I think that this is a paper of interest and is generally well written. I enjoyed reading it and I like how the latent phenotype and LPE were discussed and defined.

We thank the reviewer for their interest and positive comments.

However, I have struggled to understand the experimental set-up and rationale for using the DZ strain for quantifying phages after the 24 hour "host quality experiment"....that is if I have interpreted the experimental set up correctly. Please see my comments on this below. Until I could get some clarity on the experimental setup and the rationale behind it is hard to properly evaluate the results and discussion.

We apologize that we failed to explain the rationale for the choice of *M. xanthus* strain DZ1 to quantify the number of viable Mx1 phage particles after Mx1 growth on MyxoEE-3 populations, as well as for any other aspects of our experiment descriptions that may not have been clear. We have revised the manuscript to include this rationale for using DZ1 as an indicator strain for phage quantification and to enhance the clarity of relevant experiment descriptions. DZ1 was chosen for the plaque assay because it was previously known to be extremely susceptible to lysis by Mx1, which readily forms plaques within lawns of DZ1. We use well-established plaque assays for our quantification of Mx1 phage (the Double Agar Overlay method, Kropinski *et al.* 2009), as elaborated further below and in our Methods section.

Kropinski A.M. *et al.* (2009) Enumeration of Bacteriophages by Double Agar Overlay Plaque Assay. In: Clokie M.R., Kropinski A.M. (eds) Bacteriophages. Methods in Molecular Biology™, vol 501. Humana Press.

In line 90-91... you define host quality as the degree to which a host genotype or population facilitates parasite growth. Am I correct in thinking that for phage quantification following your 24 hour phage propagation/host quality experiment that you used strain DZ1 as the only host to quantify the phages? If this is the case then I think that there may be a fundamental problem with this approach due to efficiency of plaquing (EOP) which can vary widely between bacterial hosts. With EOP you can get huge variance in phage titres (by many orders of magnitude) for a single phage stock when tested on highly related hosts.

Yes, only DZ1 was used for the phage-quantification step at the end of the assay and we now state this more explicitly in the Methods. We define host quality as the degree to which a host population supports phage growth *in liquid*. Importantly, we are not quantifying titres resulting from growth on DZ1. In liquid and DZ1 lawns serve merely to allow us to estimate the number of viable phage particles after growth in liquid on distinct MyxoEE-3 populations. With our approach, the variance in phage numbers comes solely from differences in Mx1 growth on distinct MyxoEE-3 populations in liquid. It is precisely because we use only one phage genotype and only one *M. xanthus* plaque-permissive strain - DZ1 - that we can quantitatively compare how well Mx1 grows on the different MyxoEE-3 populations and ancestors during the liquid phase.

For clarity, we here briefly summarize the assay again. (Apologies for repeating points already understood by the referee, but we err on the side of thoroughness to insure clarity.)

First, we distinguish between two major phases of our assays to quantify the quality of MyxoEE-3 populations as Mx1 hosts, namely i) the growth phase in liquid, which is the context of host quality in this study and ii) the post-growth quantification phase, in which we determine the number of viable phage particles resulting from growth in the liquid phase.

Growth phase - growth of a clonal Mx1 stock on diverse MyxoEE-3 bacterial populations

A single phage stock was used as the common source for all of our Mx1 growth assays. Importantly, Mx1 was not part of the MyxoEE-3 evolution experiment, in which only *M. xanthus* evolved.

To initiate the host-quality growth assays, aliquots of our Mx1 source stock of identical volume and Mx1 density ($\sim 2 \times 10^6$ particles/ml) were added to cultures of different MyxoEE-3 populations (or their ancestors) of standardised bacterial density ($\sim 2 \times 10^8$ cells/ml) at T0. These bacteria-phage communities in liquid culture were then incubated for 24 h under the specified conditions to assess host quality.

Quantification phase - dilution-plating of post-growth Mx1 populations and counting the resulting plaques

Importantly, the Mx1 quantification phase was not part of the context in which host-quality was assessed. We were only interested in host quality during the liquid phase. The Mx1 quantification phase is simply an approach to quantify the size of Mx1 populations after the 24 h liquid growth phase. First, the phage were separated from the MyxoEE-3 bacteria by the specified protocol. The separated phage then underwent serial dilution followed by mixture with aliquots of strain DZ1 and these mixtures were then embedded within soft-agar overlaid on hard-agar nutrient medium. Within the soft-agar overlay, DZ1 grows into a thick lawn in the absence of Mx1. When Mx1 is present, an individual particle will grow by lysing DZ1 cells until a round, clear plaque is formed. Each plaque represents a single phage particle that was present at the end of the growth phase. By counting the plaques from a plate with an

appropriate number and multiplying by the relevant dilution factor, the total number of phage present after the 24 h of growth on an MyxoEE-3 population in liquid could be calculated. As DZ1 is a highly susceptible indicator strain, the number of phage able to form plaques on a DZ1 lawn is close to the total number of viable phage. Most importantly, using DZ1 as the indicator strain allows for the comparison between the different phage samples amplified in the MyxoEE-3 populations.

To summarize, separate samples of single phage stock are inoculated into different bacterial cultures, allowed to grow as they are able on various MyxoEE-3 cultures for 24 h in liquid and then their respective populations sizes are assessed by a plaque-number assay using DZ1, an *M. xanthus* genotype highly susceptible to lysis by Mx1.

For this reason you need to quantify phage on the host that you are interested in assessing for host quality not on a “common garden” host e.g. DZ2. By using DZ2 for quantification of the phage populations size on a given host (as a proxy for host quality) you do not have a measure of host quality on that host, you only have a measure of how it titres on DZ2 – i.e. you could have some phages that grow really well on their own host but not on DZ2 and vice versa. I hope this point makes sense as it is key to the experiment but please correct me if I have misinterpreted the design.

Actually, in the context of our experiment the opposite is correct. The different MyxoEE-3 populations cannot be used for both the phage 24-h growth assays in liquid (i.e. the host quality assays) and post-growth plaque-count assays for precisely the reason raised by the reviewer, because Mx1 might have different EOPs across different MyxoEE-3 populations and thus we would not have an identical method for quantifying viable phage particles after the 24 h liquid assays. Because we want to assess host quality in liquid culture, it is in fact very important that we use only one highly susceptible *M. xanthus* genotype for the plaque counts to avoid confounding variation in EOP. We modified the Methods to clarify this point.

In addition to this, another key issue for me is the use of the 24 hour phage propagation experiment as a measure of host quality. 24 hours is a long time from a bacterial and phage perspective and you could get selection for resistance amongst other things during this time frame. I am not sure that it is a good and accurate measure of host quality as you define it.

In the context of this study, we define host quality as simply the degree to which a host population facilitates parasite growth during 24 h in liquid culture.

24 hours is in fact a very short time with regard to the potential for resistance to Mx1 to appear and rise to appreciable frequency. Our *M. xanthus* cultures started the growth assays already at $\sim 2 \times 10^8$ cells/ml when we added the phage for the growth assays. In the medium used, *M. xanthus* can only grow to a maximum density of $\sim 2 \times 10^9$ cells/ml and does so within ~ 12 hours in the absence of phage.

M. xanthus grows slowly compared to most model-system bacteria with a doubling time of ~ 4 hrs under our experimental conditions (Velicer *et al.* 1998, *PNAS*). This means that even if an Mx1-resistant mutant of *M. xanthus* were already present among the $\sim 1.6 \times 10^8$ cells (8 ml of culture at $\sim 2 \times 10^8$ cells/ml) to which Mx1 was added to initiate each Mx1-growth assays, that mutant could only grow to a maximum of ~ 100 cells within the 24 hours of the assay, a negligible number relative to the initial bacterial population size. Moreover, even a number of ~ 100 cells could be reached by the mutant in 24 h only if the total population was prevented by phage from reaching stationary phase and thereby ceasing all population growth, including that of the mutant. (Stationary phase would normally be reached in ~ 12 h from the starting density of $\sim 2 \times 10^8$ cells/ml in the absence of phage.) Thus, we argue that it is not quantitatively plausible that Mx1-resistant mutants that first appear in the cultures used for our growth assay could have significantly impacted our results.

[However, we would note that even if this were a realistic scenario, differences in the rate at which different MyxoEE-3 populations mutate to Mx1-resistance would be fascinating and would fall within the scope of our analysis. We are interested in the relative degree to which different MyxoEE-3 populations support Mx1 growth, *irrespective of the precise mechanisms that generate such differences*. That being said, given what we know about *M. xanthus* growth, we reiterate that we think it is implausible that differences between MyxoEE-3 populations in rates of mutation to Mx1-resistance

during our growth assays might explain the differences we observe in Mx1 population sizes after growth on different MyxoEE-3 populations.]

Velicer, G. J., Kroos, L. & Lenski, R. E. (1998) Loss of social behaviors by *Myxococcus xanthus* during evolution in an unstructured habitat. *PNAS USA* 95, 12376–12380.

Perhaps a one more useful measure(s) of host quality would be to take the hosts you are interested in and doing a one-step growth curve with them... I think this would provide a much better measure of host quality as it you will get information on phage adsorption, length of infection cycle and burst size which will allow you measure the number of progeny produced per host and over what time-scale. I think this would really bolster your study as it would remove any ambiguity or alternative explanation as to why you see lower populations on different evolved hosts.

We agree with the reviewer that descriptions of more detailed components of Mx1 growth on various MyxoEE-3 populations in liquid would be interesting. However, these sub-components of phage growth are not necessary for our clearly defined measure of host quality and thus are not necessary for any of the inferences we make in this study. It could be interesting to see more detailed descriptions of Mx1 growth components pursued in a future study. We specify the focus of this study early on, namely, comparing how well a single phage genotype grows on ancestral vs evolved *M. xanthus* host genotypes after the evolved hosts underwent evolution during MyxoEE-3 without interacting with the focal phage genotype.

Another issue was with the initial experimental set-up that I am not sure I understood properly – were six different ancestral variants used to initiate the experiment? Line 221

Yes, as described in our manuscript (see Table S1), six ancestral *M. xanthus* genotypes were used to establish the MyxoEE-3 evolution experiment. Among these six, there were three distinct motility genotypes that were each represented by sister clones of two distinct marker states - rifampicin-sensitive and rifampicin-resistant. GJV1 and GJV2 were each ancestral to 36 of the MyxoEE-3 populations examined here, which represent the bulk of our study (Figs. 1-3, S1-S3). Additionally, GJV3-GJV6 were each the ancestors of 8 of the MyxoEE-3 populations, which were only examined in assays reported in Fig. 4.

*Also, Line 104 onwards – you discuss the potential receptors sites of phages but do you have any detailed information on how the phage used in this study infects *M. xanthus* and how resistance evolves – this might help inform the data that you have observed for certain selective environments? For example you investigate the *M. xanthus* motility systems – is this because phage bind to flagella or pili or what was the motivation for using these variants in the analysis.*

The motivation for examining MyxoEE-3 populations descended from the motility mutants was an interest in the possibility of i) immediate effects of removing one motility system or the other on host quality, especially with respect to removal of Type-4-pili-based motility and ii) historical genotype effects on the evolution of latent of host quality, particularly with respect to variable motility genotypes. Prior to this study it was unknown whether Mx1 phage interact with Type 4 pili. However, this was a plausible possibility given known interactions between phage and Type 4 pili in other species. In our case, we find that production of pili does not promote phage infection but rather hinders it, presumably via production of exopolysaccharides, which is known to be positively regulated by pilin production.